# Estradiol Regulates the Expression and Secretion of Antimicrobial Peptide S100A7 via the ERK1/2-Signaling Pathway in Goat Mammary Epithelial Cells

**DOI:** 10.3390/ani12223077

**Published:** 2022-11-08

**Authors:** Yutong Yan, Yuwei Niu, Yingwan Ma, Xiaoe Zhao, Menghao Pan, Baohua Ma, Qiang Wei

**Affiliations:** 1Key Laboratory of Animal Biotechnology of the Ministry of Agriculture, Northwest A&F University, Xianyang 712100, China; 2College of Veterinary Medicine, Northwest A&F University, Xianyang 712100, China

**Keywords:** S100A7, mammary epithelial cell, estradiol(E2), ERK1/2, goat

## Abstract

**Simple Summary:**

S100A7 plays an essential role in goat mammary glands for a broad spectrum of resisting the invasion and infection of pathogenic microorganisms, and also is considered an alternative strategy for preventing and treating mastitis to broad-spectrum antibacterial strategies, with almost no resistance. As we know, estrogen also participates in the body’s immune functions, acting either as pro-inflammatory or anti-inflammatory mediators, yet whether estrogen is involved in the expression and secretion of S100A7 remains unclear. In the present study, the relationship between estrogen and S100A7 concentration in goat milk was positive, and the regression equation was y = 0.3206x + 23.459; then the goat mammary epithelial cells (gMECs) were isolated and treated with estradiol; the results showed that the expression and secretion of S100A7 were induced with estradiol treatment via the ERK1/2-signaling pathway, and both nuclear and membrane receptors participated in this process. This study provides evidence that estradiol induces the expression and secretion of antimicrobial peptide S100A7 in gMECs through the ERK1/2-signaling pathway, and lays a theoretical foundation for exploring the phenomenon that the incidence of clinical mastitis is highest during the dry period and perinatal period, and the role of estradiol in regulating the synthesis and secretion of S100A7 in gMECs.

**Abstract:**

S100A7 has received extensive attention in the prevention and treatment of mastitis across a broad spectrum, yet there is a little information about its mechanism, especially in the immunomodulatory effects of estrogen. In the present study, based on the milk bacteriological culture (BC) of 30 dairy goats, the concentration of both estrogen and S100A7 in the BC-positive samples was not significantly different than in the BC-negative samples; the estrogen abundance in subclinical and clinical mastitis samples also showed only a limited difference; compared with healthy samples, the S100A7 abundance in subclinical mastitis samples differed little, while it was significantly decreased in clinical mastitis samples. Moreover, the relationship between estrogen and S100A7 was positive, and the regression equation was y = 0.3206x + 23.459. The goat mammary epithelial cells (gMECs) were isolated and treated with 1, 10, 100 nM E2 and/or 5 μg/mL lipopolysaccharide (LPS), respectively, for 6 h. Compared with control samples, 5 μg/mL LPS, 10 nM E2 and 100 nM E2 markedly induced S100A7 expression and secretion. More than separated treatment, the cooperation of LPS and E2 also significantly increased S100A7 expression, rather than S100A7 secretion. The p-ERK was up-regulated markedly with 100 nM E2 treatment, while the expression of p-JNK, p-p38 and p-Akt had little effect. The G protein-coupled estrogen receptor 1(GPER1) agonist G1 markedly induced S100A7 expression and secretion in gMECs, and the estrogen nuclear receptor antagonist ICI and GPER1 antagonist G15 significantly repressed this process. In conclusion, E2 binds to nuclear and membrane receptors to regulate the expression and secretion of S100A7 via the ERK1/2-signaling pathway in gMECs.

## 1. Introduction

Mastitis is one of the most common and serious diseases in dairy animals [1], and is generally caused by bacterial infection of the mammary gland, which leads to countless economic losses all over the world [2,3,4]. A higher incidence of clinical mastitis in the perinatal period and dry milk period is common in goats and cows [5]. In the perinatal period, parturition has been shown to induce changes in the blood neutrophil gene expression in dairy cows [6,7], so it is easy to explain the dysfunctional capacity of leukocytes in perinatal cows for the repressed expression of these genes [8]. This suggests that the gene expression of neutrophil may be related to a change in serum P or E2 [7], yet it is an undisputed fact that the two hormones decline during the perinatal period in goats and cows. Postpartum goats or cows are weak and their immune function is decreased, and the priority of scarce metabolizable protein allocation to milk production over immune function also cannot be ignored, although it may be gradual rather than absolute [9]. For this reason, the higher incidence of mastitis in the perinatal period and dry milk period is worth exploring and finding out the reasons. Notably, estrogen plays an essential role in body immune function as well as delivery. Estradiol (E2), a predominant and the most potent sexual hormone during the reproductive stage in females [10], is mainly associated with reproduction; it regulates the proliferation, differentiation and survival [10], and is also involved in innate immune response (IIR) [11,12]. During the middle period of gestation, the level of E2 rises abruptly before parturition and peaks in late pregnancy before delivery, then falls rapidly and regains the basal level [13,14,15]. On the other hand, the high incidence of mastitis occurs both before and after delivery; yet the role of E2 in IIR remains unclear, especially its role of antimicrobial peptide (AMP) expression.

S100A7, an AMP with a strong antimicrobial activity [16,17], which is first identified in the epithelial cells of human psoriasis skin [18,19], has been proved to play a critical role in IRR against the invasion and infection of pathogenic microorganisms. S100A7 can be detected in bovine milk [20,21] and serum [22]. In a bovine udder, the expression of S100A7 is common in teat skin and the streak canal [20,21]. But the expression of S100A7 was limited to bacteria-infected teat cisterns rather than the epithelial cells of the alveolus [17,23]. Conversely, in goats, less information exists about the expression and regulation mechanism of S100A7 than for bovines; it is reported that S100A7 is expressed in the stratified squamous epithelium of the teat, epithelial cells of the alveolus and gland cistern in healthy goats, that lipopolysaccharide (LPS) induced the expression and secretion of S100A7 in goat mammary epithelial cells [24] and that S100A7 concentration was increased with intramammary infusions of LPS for 24 h [25]. Defensins, belonging to AMPs, have been proved to be induced with estrogen or estradiol in human vaginal epithelial cells [26] and in sheep oviductal epithelial cells [27]. Nevertheless, there is little known about the relationship between S100A7 and E2, especially in goat milk and goat mammary epithelial cells (gMECs). As we know, gMECs participate in the innate immune response of producing S100A7 in the goat mammary gland [24,25], but the role of E2 on the response is still unclear, and whether alveolus epithelial cells can be induced in S100A7 synthesis and secretion by E2 to participate in innate immunity in the mammary gland remains poorly understood.

In order to investigate the role of estrogen in regulating the expression and secretion of S100A7 in goat MECs, S100A7 and estrogen abundance in dairy goat milk were detected and their relationship analyzed, and gMECs were isolated and treated in vitro with E2 to explore their connection. Results indicated that there was a positive relationship between S100A7 and estrogen in milk, that E2 could promote the expression and secretion of S100A7 in gMECs, that this promotion effect was dependent on the ERK1/2-signaling pathway, and that both estrogen nuclear and membrane receptors were involved in this process.

## 2. Materials and Methods

### 2.1. Primary Antibodies and Chemicals

All chemicals were purchased from Sigma–Aldrich, unless otherwise specified. As for 17β-estradiol (E2758), GPER1 agonist G1 (10,008,933), GPER1 antagonist G15 (14,673) and ICI182780 (1,011,269), these were purchased from Cayman. ERK1/2 antibody (4695) and phosphor-ERK1/2 antibody (4370) were purchased from Cell Signaling Technology; Akt antibody (BM4930), phosphor-Akt antibody (BM4838), β-actin antibody (BM0627) were purchased from BOSTER; p38 antibody (sc-7972), phosphor-p38 antibody (sc-7973), JNK antibody (sc-7345), phosphor-JNK antibody (sc-6254), β-casein (sc-166530) and CK-14 (sc-53253) antibody were purchased from Santa Cruz Biotechnology (Santa Cruz, CA, USA); CK-18 antibody (Abcam, Cambridge, MA, USA, ab668).

### 2.2. Milk Collection and Mammary Epithelial Cells Isolation and Culture

Half-udder milk of 30 dairy goats was sampled at peak lactation in the Yang ling area of Shannxi province, China. Dairy goat mammary tissue samples (*n* = 12) were collected from a slaughterhouse in Shaanxi, China, the dairy goats had no evident clinical signs of clinical mastitis and milk bacteriological culture was negative. Milk was collected with sterile tubes after disinfecting it three times with 75% alcohol and discarding the first three handfuls milk; it was transported at low temperature; milk bacteriological culture was performed as recommended previously. Briefly, milk samples were hand-mixed and opened in a biosafety level II cabinet, 10 µL milk was streaked using the quadrant streaking method over an agar base, plates were incubated at 37 °C, and then read after 24 h to 48 h. In case no growth was visible after the first 24 to 48 h of incubation, plates were reincubated (37 °C) and rechecked at 24 h intervals for up to 96 h. If the culture of milk was negative and the goats had no evident signs of clinical mastitis, the glands were sampled. Isolation and culture of gMECs were processed as previously described [24]. Briefly, the tissues were washed with PBS and minced into about 1 mm^3^ cubes, then implanted into the 35 mm Petri dish and incubated at 37 °C with saturated humidity and 5% CO_2_, then 1.5 mL culture medium was added to the culture dish and replaced with fresh medium every 48  h, finally, the cells spread across the bottom of the dish were passaged by digestion with 0.25% trypsin/EDTA(Invitrogen Corporation, Waltham, MA, USA, 15,090,046) and were reseeded. In these experiments, cells passaged within 8 times were used.

### 2.3. Immunofluorescence

Cells were fixed for 30 min with 4% paraformaldehyde (Solarbio, Beijing, China P1110) and permeabilized for 10 min with 0.2% TritonX-100 (Sigma, St. Louis, Mosby, MO, USA, T8787), then incubated for 2 h in a blocking buffer (3% BSA in PBS) and treated with primary antibody against CK18, CK14 overnight at 4 °C. The secondary antibody used was goat anti-Rabbit IgG-Alexa Fluor^®^ 488 (Abcam, Cambridge, MA, USA, ab150077). The nuclei were stained with DAPI (Beyotime, Shanghai, China, C0060) for 5 min at room temperature. Images were captured using a fluorescence microscope (Olympus, Tokyo, Japan, IX71).

### 2.4. RNA Reverse Transcription and Polymerase Chain Reaction

Total RNA was extracted using RNAiso Plus (Takara, Dalian, China, 19,109), according to the manufacturer’s instruction. cDNA was synthesized with PrimeScript™ RT Reagent Kit (Takara, Dalian, China, RR036). The primer sequences and data analysis methods were cited in the previous study [24]. The quantitative RT–PCR reactions were performed using the Fast SYBR Green Master Mix (Genstar, Beijing, China, A302-01); data collection and data analysis were performed on the QuantStudio 6 Flex machine (Invitrogen Corporation, Waltham, MA, USA) by using GraphPad Prism 6 software. The quantitative RT–PCR parameters were as follows: 95 °C for 2 min, followed by 40 cycles each at 95 °C for 15 s, 60 °C for 30 s, and 72 °C for 30 s.

### 2.5. Western Blot

Cells were lysed using the RIPA buffer (Solarbio, Beijing, China R0010) and supplemented with 1 mM phenylmethylsulfonyl fluoride (Solarbio, Beijing, China P0100) on ice for 30 min. Western blot was carried out with equal amounts of protein. The primary antibodies were incubated overnight at 4 °C, then the corresponding secondary antibodies were incubated at room temperature, and the protein bands were detected with a chemiluminescence kit (Biotanon, Shanghai, China, tanon5200).

### 2.6. ELISA

S100A7 and estrogen abundance was measured using ELISA. ELISA was performed using Goat S100A7 and the estrogen ELISA kit according to the manufacturer’s protocol. The limits of quantification of the Goat S100A7 ELISA kit (Rui Xing, Quanzhou, China, RXJ1100386G) we used are 2.5–80 μg/mL, the sensibility is less than 0.1 μg/mL; The limits of quantification of the Goat estrogen ELISA kit (Rui Xing, Quanzhou, China, RX1100403G) we used are 3.2–80 pg/mL, the sensibility is less than 1 pg/mL. The absorbance was measured at 450 nm using the microplate reader (Tecan).

### 2.7. Statistics

All quantitative data are presented as the mean ±  standard deviation (SD) with no fewer than three replicates for each experimental condition. An unpaired Student’s test was used to compare the two groups; one-way ANOVA was used to compare multiple groups using SPSS 17.0. A value of *p* > 0.05 was considered to constitute no significant difference; a value of *p* < 0.05 was considered significant difference; a value of *p* < 0.01 was considered extremely significant difference.

## 3. Result

### 3.1. Difference between Estrogen and S100A7 Abundance in Milk and Their Correlation

Based on the bacteriological culture of milk, 30 dairy goats were divided into a BC-negative group (*n* = 15) and a BC-positive group (*n* = 15). The abundance of estrogen and S100A7 are shown in Figure 1. Both estrogen and S100A7 in milk showed no significant difference between the BC-positive group and the BC-negative group (Figure 1A,B, *p* > 0.05). The BC-positive samples were further divided based on the clinical symptoms of mastitis; the estrogen abundance in the three groups had a limited significance (Figure 1C, *p* > 0.05). Notably, the S100A7 abundance in subclinical mastitis samples had little difference compared with healthy samples (Figure 1D, *p* > 0.05) while there was a significant decrease in clinical mastitis samples (Figure 1D, *p* < 0.01), and there was also a significant difference between subclinical and clinical mastitis samples (Figure 1D, *p* < 0.05). Moreover, the correlation between estrogen and S100A7 was positive, and the regression equation was y = 0.3206x + 23.459 (Figure 2).

### 3.2. Isolation and Identification of gMECs

In order to further analyze the immunomodulatory effects of estrogen in goat mammary glands, gMECs were isolated to explore the regulatory mechanism of S100A7 expression with E2 treatment. After about 5 days culture, cells migrated out of the mammary tissue (Figure 3A) and then spread across the bottom of the dish after another 5 days (Figure 3B). The cells were dissociated using 0.25% trypsin/EDTA and reseeded in a new 35 mm Petri dish. The isolated cells possessed typical epithelial cell morphology, including colony forming and cobblestone-like shapes (Figure 3C). If the isolated cells were cultured at a high density, a dome-like structure could be observed (Figure 3C). Immunofluorescence results showed that the isolated cells expressed cytokeratin typical of differentiated luminal epithelial cells (CK18) (Figure 3D) but not myoepithelial cells (CK14) (Figure 3E). Furthermore, the protein of β-casein was detected using western blot(Figure 3F) in isolated cells. These results indicate that the isolated cells were goat mammary epithelial cells.

### 3.3. E2 Induced the mRNA Expression and Secretion of S100A7 in Cultured gMECs

In order to determine whether E2 can induce the expression of S100A7 in gMECs cultured in vitro, the isolated gMECs were treated with 1 nM, 10 nM, 100 nM E2 and/or LPS (5 μg/mL), respectively, for 6 h; LPS is a positive group which has been proved to induce S100A7 mRNA expression and secretion in gMECs [24,28]. Firstly, the concentrations of LPS and E2 had no significant effect on the cell viability of gMECs for 24 h (Figure 4A, *p* > 0.05), the original data of Cell viability(OD) is shown in Appendix A, which showed that they can be used for subsequent research. For 5 μg/mL LPS, 10 nM and 100 nM E2 groups, after being treated for 6 h, an increase in S100A7 mRNA expression was detected compared to the control group (Figure 4B, *p* < 0.01), while there was a little difference in the 1 nM E2 group (Figure 4B, *p* > 0.05); the cooperation of E2 and LPS future promoted S100A7 mRNA expression rather than separated treatment (Figure 4B, *p* < 0.01), the original data of S100A7 mRNA expression of gMECs with LPS or estradiol treatment for 6 h is shown in Appendix A. S100A7 secretion in the medium was analyzed in Figure 4C, for 5 μg/mL LPS, 10 nM and 100 nM E2 groups; after being treated for 6 h, an increase in S100A7 secretion was detected compared to control groups (Figure 4C, *p* < 0.01); the 1 nM E2 group shows no significant difference compared to the control group (Figure 4C, *p* > 0.05), yet the cooperation of E2 and LPS did not further promote S100A7 secretion (Figure 4C, *p* > 0.05), the original data of S100A7 concentration in gMECs with LPS or estradiol treatment for 6 h is shown in Appendix A. These results indicated that E2 can induce the mRNA expression and secretion of S100A7 in gMECs.

### 3.4. E2-Activated ERK1/2-Signaling Pathways in Cultured gMECs

gMECs were treated with 1 nM, 10 nM and 100 nM E2 for 6 h. The protein levels of p-ERK/ERK, p-JNK/JNK, p-p38/p38 and p-AKT/AKT were tested. As the results show, the protein level of p-ERK was up-regulated after 100 nM E2 treatment for 6 h (Figure 5B, *p* < 0.01), yet the 1 nM and 10 nM E2 groups showed no significant difference (Figure 5B, *p* > 0.05). For the three groups, different concentrations of E2 also had a limited influence on the protein levels of p-JNK/JNK, p-p38/p38 and p-AKT/AKT (Figure 5B, *p* > 0.05). The original data of relative strip gray of protein with estradiol treatment for 6 h is shown in Appendix A. These results indicate that E2-activated ERK1/2-signaling pathways induce S100A7 expression and secretion in cultured gMECs.

### 3.5. E2 Binds to Both Nuclear and Membrane Receptors to Induce Expression and Secretion of S100A7

Estrogen acts by binding to its receptor; therefore, an ICI (nuclear receptor inhibitor, 1 μM), a G15 (membrane receptor inhibitor, 1 μM) and a G1 (membrane receptor agonist, 0.1 μM) were used to explore through which receptors S100A7 was up-regulated. Compared with the control group, the expression and secretion of S100A7 in the E2 group showed a significant increase (Figure 6A,B, *p* < 0.01), which was inhibited significantly with the cooperation of ICI or G15 (Figure 6A,B, *p* < 0.01), and G1 also significantly induced S100A7 expression and secretion (Figure 6A,B, *p* < 0.01). The original data of S100A7 mRNA expression in gMECs after treatment for 6 h is shown in Appendix A; the original data of S100A7 concentration in gMECs after treatment for 6 h is shown in Appendix A. Results show that both nuclear and membrane receptors join the synthesis and secretion of S100A7 in gMECs with estradiol treatment.

## 4. Discussion

Mastitis causes huge economic losses in the milk industry all over the world [29,30], and has a high incidence during the perinatal period. Antibiotics are one of the first choices for treating mastitis, yet there are consequent disadvantages, mainly in the decline of milk quality and the emergence of drug-resistant microbes [31,32]; therefore, more attention was paid to the alternative strategies of preventing and treating mastitis [33]. Now, the antimicrobial peptide (AMP) becomes a valuable substitute for a broad spectrum of pathogenic microorganisms with almost no resistance [33].

S100A7, an antimicrobial peptide, is also considered an inflammation-related protein [34,35], which was first found in patients with psoriasis in 1991 [19]; subsequently, more and more studies have proved that S100A7 plays an essential role in innate immunity [16,17,22,24]. The expression and secretion of S100A7 is regulated by a variety of factors, such as pathogenic microorganism [16], IL-1β [16], IL-37 [36], chemokines [34], LPS [24,25] and others. It is well known that the decline of estrogen in peripheral blood not only plays a vital role in production during the perinatal period, but also participates in body immunity regulation. Nevertheless, it remains unclear whether the high incidence of mastitis is related to the decline of estrogen levels which lead to impaired immune functions. As we know, estrogens have anti-inflammatory or pro-inflammatory functions that act throughout the body [37]. Notably, the role of estrogen concerning the expression and secretion of S100A7 is still unclear, especially in goats.

In order to explore the relationship between estrogen and S100A7, their abundances in goat milk were detected first. Based on the bacteriological culture (BC) of goat milk, both estrogen and S100A7 abundance in mastitis samples showed a limited difference compared to healthy samples. Considering that the severity of inflammation varies between subclinical and clinical mastitis [38], samples were further subdivided into subclinical and clinical mastitis according to the symptoms of clinical mastitis. Interestingly, although estrogen abundance in the three groups also showed no significant difference, S100A7 abundance significantly decreased in clinical mastitis samples. Visible changes of milk and mammary glands in clinical mastitis goats were presented and easy to check [24,38]; the breast function was impaired seriously, especially in gMECs, which proved that this can induce the expression and secretion of S100A7 and collapsed alveolus of mastitis goat glands [24]; therefore, it is normal that S100A7 abundance in clinical mastitis milk was decreased, due to the serious damage of the gMECs function. Of course, the immune function of mammary gland is a complex physiological process, which can be affected by countless factors, so their abundances in milk need not directly reflect the relationship between S100A7 and estrogen, and the effect of estrogen on the synthesis and secretion of antimicrobial peptide S100A7 remains unclear, especially in goats.

As we know, goat mammary epithelial cells are the basic units of mammary gland lactation functions, which play an important role in the process of lactation and mammary gland immunity. In order to explore the role of estrogen in regulating the synthesis and secretion of S100A7, goat mammary epithelial cells were isolated, identified with CK18 [39] and their lactation function with β-casein [40]. Considering estradiol is the most important and most biologically active hormone in estrogen [10], gMECs were treated with estradiol to explore the relationship between S100A7 and estradiol. According to recent research, LPS (5 μg/mL) can induce the synthesis and secretion of S100A7 in gMECs for 6 h. LPS also induced the expression and secretion of S100A7 in the present study, which supports the previous research [24,28], so we used it as a positive group at this point in the whole experiment. gMECs were further treated with different estradiol concentrations for 6 h. Interestingly, the expression of S100A7 was significantly induced with estradiol treatment as well as S100A7 secretion, which supports the research that S100A7 was induced with estradiol treatment in bovine mammary epithelial cells [12]. S100A7 expression had a significantly higher level in the groups of combined estradiol and LPS than estradiol treatment alone, while S100A7 secretion showed only a limited difference in the groups of combined treatment and showed that estradiol could induce the expression and secretion of S100A7 in gMECs, which further supports the relationship between estrogen and S100A7 abundance in milk and E2 induced S100A7 expression as well as β-defensin 1 or β-defensin 5 in bovine mammary epithelial cells [12].

The signaling pathway through which estradiol regulates the expression and secretion of S100A7 remains unclear. Mitogen-activated protein kinases (MAPKs) signal pathway and PI3K/AKT play essential roles in the body [41,42,43], and estradiol too, via these pathways, joins various activities. Moreover, MAPKs include three major subfamilies, the c-jun N-terminal kinases (JNKs), the p38 MAPKs and the extracellularly responsive kinases (ERKs) [41,42]. The expression and secretion of S100A7 was induced in gMECs especially with C-type natriuretic peptide treatment via the JNKs signaling pathway [28]. While the question of whether estradiol participates in the synthesis and secretion of S100A7 through these signaling pathways in gMECs still remains unclear, these pathways were detected with estradiol treatment, and results show that the ERK signal pathway was activated while other pathways had little significance. This shows that the ERK signal pathway participated in the synthesis and secretion of S100A7 in gMECs with estradiol treatment.

Estrogen acts by binding to its receptor. It has been proved that bovine mammary epithelial cells express both ERα and Erβ [44], and it has been reported that both nuclear and membrane receptors were expressed in gMECs [45]; therefore, an ICI (nuclear receptor inhibitor), a G15 (membrane receptor inhibitor) and a G1 (membrane receptor agonist) were used to explore through which receptors S100A7 was up-regulated. Results showed that the synthesis and secretion of S100A7 in gMECs with estradiol treatment were suppressed when estrogen nuclear and membrane receptors were inhibited, and also were induced with G1 (GPER1 agonist), which showed that estradiol bound to its nuclear and membrane receptors to induce the synthesis and secretion of S100A7 via activating the ERK-signaling pathway in gMECs.

Estrogens play vital roles in the body’s innate immunity, by acting either as proinflammatory or anti-inflammatory mediators. In the present study, S100A7 expression and secretion can be induced with estradiol in gMECs, which shows that estradiol acted as an anti-inflammatory mediator due to the broad spectrum antibacterial activity of S100A7 [16]; therefore, the high incidence of mastitis during the perinatal period may be relative to the level of estrogens, and the other functions of S100A7 also cannot be ignored [16]. Notably, the cooperation of LPS and estradiol can to a limited extent increase S100A7 secretion, which shows that there was no direct relationship. Of course, it is different that LPS activated MYD88 signaling by binding to the TLR4 receptor in previous research [24] while estradiol activated ERK signaling by binding to the nuclear and membrane receptors of estrogen in the present study. Estrogen plays a vital role in the body’s IRR; it is not only involved in the synthesis and secretion of S100A7, but also may regulate the expression of other antimicrobial peptides.

In this paper, we investigated the role of estrogen in regulating the expression and secretion of S100A7 in goat MECs. The results showed that the S100A7 level is positively correlated with estrogen in goat milk, and the expression and secretion of S100A7 was up-regulated with estradiol treatment in goat MECs. This regulatory process was realized through the activation of the ERK-signaling pathway. Collectively, estrogen activated the ERK-signaling pathway to induce the expression and secretion of antimicrobial peptide S100A7, which had certain significance for understanding the specific role of estrogen in regulating the expression and secretion of antimicrobial peptides S100A7.

## 5. Conclusions

In summary, both antimicrobial peptide S100A7 and estrogen abundance were detected in dairy goat milk; their abundances had a limited significance in healthy and mastitis goat milk, and their relationship was positive; the regression equation was y = 0.3206x + 23.459. The expression and secretion of S100A7 was up-regulated with estradiol treatment in gMECs; estradiol induced the expression and secretion of S100A7 in gMECs dependent on the ERK signal pathway. Moreover, both ER and GPER1 join the expression and secretion of S100A7 in gMECs.

## Figures and Tables

**Figure 1 animals-12-03077-f001:**
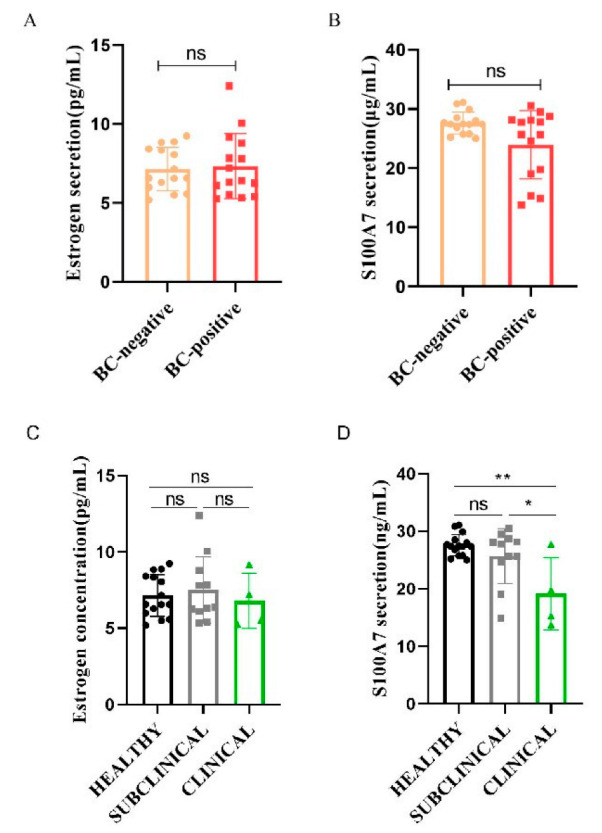
Difference between estrogen and S100A7 abundance in milk. (**A**) The difference of estrogen abundance in the BC-negative and BC-positive groups; (**B**) the difference of S100A7 abundance in the BC-negative and BC-positive groups; (**C**) the difference of estrogen abundance in HEALTHY, SUBCLINICAL and CLINICAL groups; (**D**) the difference of estrogen abundance in HEALTHY, SUBCLINICAL and CLINICAL groups; BC = bacterial culture; HEALTHY = healthy samples; SUBCLINICAL = subclinical mastitis samples; CLINICAL = clinical mastitis samples; ns: *p* > 0.05; *: *p* <  0.05; **: *p*  <  0.01. Different colors stand for different groups.

**Figure 2 animals-12-03077-f002:**
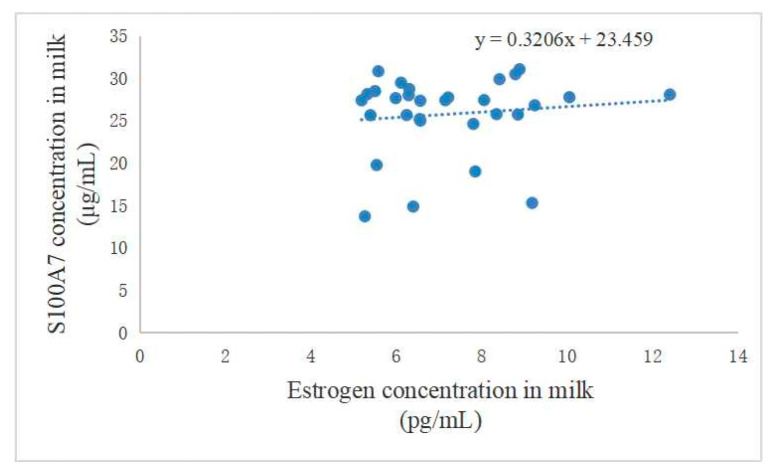
The relationship between estrogen and S100A7 abundance in milk.

**Figure 3 animals-12-03077-f003:**
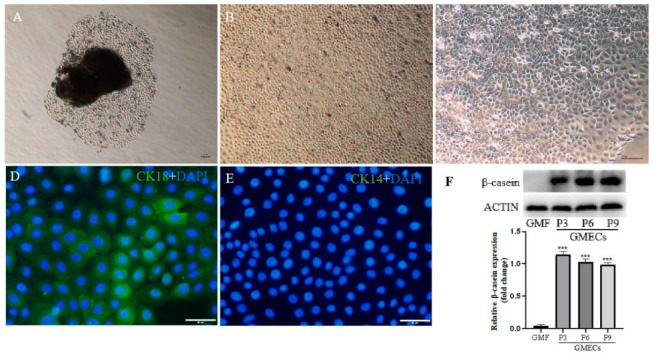
Characterization of gMECs. (**A**) Representative images of gMECs that migrated out of the mammary tissue after about 5 days’ culture; (**B**) representative images of gMECs spread across the bottom of the dish after another 5 days’ culture; (**C**) dome-like structures could be observed in cells cultured at a high density; (**D**) representative images of cytokeratin 18 (CK-18) detected using immunofluorescence; (**E**) representative images of cytokeratin 14 (CK-14) detected using immunofluorescence; (**F**) the β-casein of gMECs expressed in passages 3, 6 and 9 and was detected by western blot; GMF = goat mammary fibroblast; GMECs = goat mammary epithelial cells; ***: *p*  <  0.001. Original western blot figures in Appendix A.

**Figure 4 animals-12-03077-f004:**
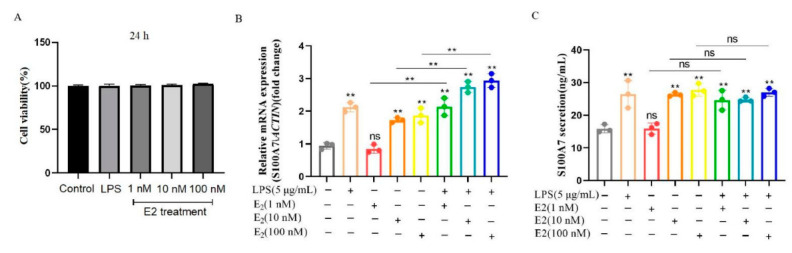
E2 induced mRNA expression and secretion of S100A7 in cultured gMECs. (**A**) Cell viability of gMECs with LPS or estradiol treatment for 24 h; (**B**) S100A7 expression of gMECs with LPS or estradiol treatment for 6 h; (**C**) S100A7 secretion of gMECs with LPS or estradiol treatment for 6 h; ns: *p* > 0.05; **: *p*  <  0.01. Different colors stand for different groups. The cell viability(OD) for (**A**) was showed in Appendix A. The level of S100A7 mRNA expression for (**B**) was showed in Appendix A. The concentration of S100A7 secretion in gMECs for (**C**) was showed in Appendix A.

**Figure 5 animals-12-03077-f005:**
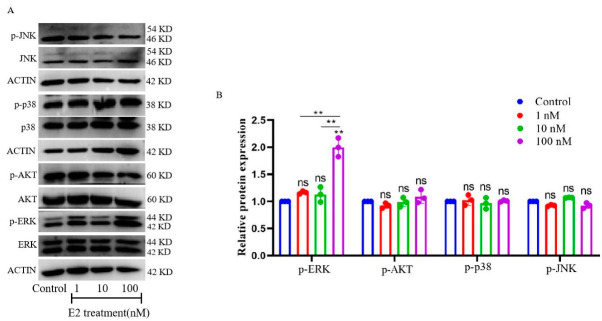
E2-activated ERK1/2-signaling pathway in cultured gMECs. (**A**) The relative proteins of gMECs were detected using western blot with estradiol treatment for 6 h; (**B**) the related gray stripe was analyzed; ns: *p* > 0.05; **: *p*  <  0.01. Different colors stand for different groups. The relative strip gray of protein with estradiol treatment in gMECs for (**B**) was showed in Appendix A. Original western blot figures in Appendix A.

**Figure 6 animals-12-03077-f006:**
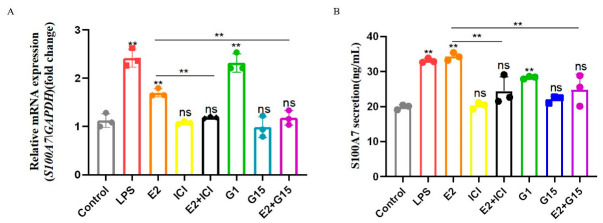
E2 binds to both nuclear and membrane receptors to induce expression and secretion of S100A7. (**A**) S100A7 expression of gMECs after treatment for 6 h; (**B**) S100A7 secretion of gMECs after treatment for 6 h; ICI (estrogen nuclear inhibitor, 1 μM); G15 (estrogen membrane inhibitor, 1 μM); G1 (estrogen membrane agonist, 0.1 μM); ns: *p* > 0.05; **: *p*  <  0.01. Different colors stand for different groups. The level of S100A7 mRNA expression for (**A**) was showed in Appendix A. The concentration of S100A7 secretion in gMECs for (**B**) was showed in Appendix A.

## Data Availability

Not applicable.

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
