# Peer review of "Estradiol Regulates the Expression and Secretion of Antimicrobial Peptide S100A7 via the ERK1/2-Signaling Pathway in Goat Mammary Epithelial Cells"

_animals, 2022, doi:10.3390/ani12223077_

Round 1

Reviewer 1 Report

Introduction

-The incidence of clinical mastitis is highest during the dry period and perinatal period, especially in the perinatal period, body immune function was impaired, which leads to the increased incidence and severity of mastitis in dairy cows:

The authors must explain the background to this statement.

-The objectives of the study should be clearly described.

Materials and methods

2.2. The procedure for selection of animals from which mammary glands were collected must be described.

2.7. With regard to the replicates, what analysis was performed, and what results were taken into comparison?

Results

Figures 4, 5 and 6 are OK, but the authors must provide more detailed results in tables, which should be inserted as supplementary material.

Discussion

First, there are relevant references with work in goats, which were published in the 1980’s and 1990’s, which should be cited.

Second, the discussion must be divided in subsections for easier flow of reading.

Third, the authors must add a new paragraph with the clinical implications of this study.

Author Response

We would like to thank the reviewer for their accurate and detailed revision of our manuscript. Reviewer offered helpful and constructive suggestions and as a result, we feel that the manuscript has been significantly improved. We have seriously thought about the suggestions and provided our responses in a point-by-point manner.

Introduction

The incidence of clinical mastitis is highest during the dry period and perinatal period, especially in the perinatal period, body immune function was impaired, which leads to the increased incidence and severity of mastitis in dairy cows:

Point 1: The authors must explain the background to this statement.

Response 1: Thanks for reviewer’s kindly reminding and advices, and we describe it clearly in line 47-57. A higher incidence of clinical mastitis in the perinatal period and dry milk period is common in goats and cows[1]. In the perinatal period, parturition has been shown to induce changes in blood neutrophil gene expression in dairy cows[2,3], so it’s easy to explain the dysfunctional capacity of leukocytes from perinatal cows for the repressed expression of these genes[4], which suggested that the gene expression of neutrophil may be related to change in serum P or E2[3], yet it’s an undisputed fact that the two hormones decline during the perinatal period in goats and cows. Postpartum goats or cows are weak and immune function is decreased, and the priority of scarce metabolizable protein allocation to milk production over immune function also can’t be ignored, although it may be gradual rather than absolute[5], so the higher incidence of mastitis in the perinatal period and dry milk period is worth exploring and finding out the reasons.

Point 2: The objectives of the study should be clearly described.

Response 2: Thanks for reviewer’s advices, and we have revised in this manuscript. In this manuscript, we investigated the role of estrogen in regulating the expression and secretion of S100A7 in goat MECs, which was described in line 85 for “In order to investigate the role of estrogen in regulating the expression and secretion of S100A7 in goat MECs”.

Materials and methods

Point 3: 2.2. The procedure for selection of animals from which mammary glands were collected must be described.

Response 3: Thanks for reviewer’s kindly reminding and advices, and we have revised in line 106-115. Before collecting goat mammary tissue, the dairy goats were detected and had no evident clinical signs of clinical mastitis (udders were swollen and hard, there are floccules, clots or yellow color in the milk, even stop lactation), then milk was sampled by sterile tube after disinfecting three times with 75% alcohol and discarding the first three handfuls milk, transportation with low temperature; milk bacteriological culture was performed as recommended previously. Briefly, milk samples were hand-mixed and opened in a biosafety level II cabinet, 10 µL milk was streaked by the quadrant streaking method over agar base, plates were incubated at 37 ℃, and then read after 24 h to 48 h. In case no growth was visible after the first 24 to 48 h of incubation, plates were reincubated (37 °C) and rechecked at 24 h intervals for up to 96 h. If the culture of milk was negative and the goats had no evident signs of clinical mastitis, the glands were sampled.

Point 4: 2.7. With regard to the replicates, what analysis was performed, and what results were taken into comparison?

Response 4: Thanks for reviewer’s kindly reminding and advices. Unpaired Student’s test was used to compare between two groups, one-way ANOVA was used to compare among multiple groups using the SPSS 17.0, which was revised in line 156-158.

Results

Point 5: Figures 4, 5 and 6 are OK, but the authors must provide more detailed results in tables, which should be inserted as supplementary material.

Response 5: Thanks for reviewer’s kindly reminding and advices. We provided more detailed results in supplementary materials(Supplemental table 1-6).

Discussion

Point 6: First, there are relevant references with work in goats, which were published in the 1980’s and 1990’s, which should be cited.

Response 6: Thanks for reviewer’s advices, we have cited the reference “Economic losses from and the national research program on mastitis in the United States.” and “Mastitis--progress on control” in line 260, which were more relevant to the work.

Point 7: Second, the discussion must be divided in subsections for easier flow of reading.

Response 7: Thanks for reviewer’s kindly reminding and advices. We divided the discussion from five subsections into eight subsections, the subsections are clearer and logical.

Point 8: Third, the authors must add a new paragraph with the clinical implications of this study.

Response 8: Thanks for reviewer’s advices. “In this manuscript, we investigated the role of estrogen in regulating the expression and secretion of S100A7 in goat MECs. The results showed that the S100A7 level is positively correlated with estrogen in goat milk, and the expression and secretion of S100A7 was up-regulated with estradiol treatment in goat MECs. This regulatory process was realized through activation of ERK signaling pathway. Collectively, estrogen activated ERK signaling pathway to induce the expression and secretion of antimicrobial peptide S100A7, which had certain significance for understanding the specific role of estrogen in regulating the expression and secretion of antimicrobial peptides S100A7.” were added in the last paragraph(line 350-358).

Supplementary material

1The cell viability(OD) for figure 4A was showed in table 1.

Table 1. Cell viability(OD) of gMECs with LPS or estradiol treatment for 24 h

Replicates

Control

LPS(5 μg/mL)

1 nM E2

10 nM E2

100 nM E2

Replicate 1

0.381

0.381

0.385

0.386

0.393

Replicate 2

0.388

0.392

0.381

0.392

0.389

Replicate 3

0.381

0.377

0.408

0.383

0.395

  • The level of S100A7 mRNA expression for figure 4B was showed in table 2.

    Table 2. The level of S100A7 mRNA expression of gMECs with LPS or estradiol treatment for 6 h

Replicates

Control

LPS(5 μg/mL)

1 nM E2

10 nM E2

100 nM E2

LPS+ 1nM E2

LPS+ 10 nM E2

LPS+ 100 nM E2

Replicate 1

0.978

2.123

0.746

1.729

2.096

2.138

2.739

2.727

Replicate 2

1.014

2.265

0.799

1.654

1.646

1.873

2.572

2.938

Replicate 3

0.829

1.982

0.988

1.805

1.871

2.404

2.907

3.148

  • The concentration of S100A7 secretion in gMECs for figure 4C was showed in table 3.

Table 3. The concentration of S100A7 secretion in gMECs with LPS or estradiol treatment for 6 h

Replicates

Control

LPS(5 μg/mL)

1 nM E2

10 nM E2

100 nM E2

LPS+ 1nM E2

LPS+ 10 nM E2

LPS+ 100 nM E2

Replicate 1

15.39

30.64

16.23

26.17

29.75

27.54

25.54

25.8

Replicate 2

17.27

22.25

17.43

26.96

27.46

24.74

24.3

28.23

Replicate 3

14.97

26.445

14.12

25.88

25.97

21.56

24.35

27.015

  • The relative strip gray of protein with estradiol treatment in gMECs for figure 5B was showed in table 4.

Table 4. The relative strip gray of protein with estradiol treatment for 6 h

Group

Replicates

pERK/ERK

p-AKT/AKT

p-p38/p38

p-JNK/JNK

Control

Replicate 1

1

1

1

1

Replicate 2

1

1

1

1

Replicate 3

1

1

1

1

1 nM E2

Replicate 1

1.192

0.876

1.118

0.910

Replicate 2

1.137

0.970

0.928

0.937

Replicate 3

1.164

0.923

1.023

0.924

10 nM E2

Replicate 1

0.988

0.904

1.067

1.074

Replicate 2

1.265

1.069

0.861

1.065

Replicate 3

1.127

0.986

0.964

1.070

100 nM E2

Replicate 1

1.998

1.060

0.991

0.873

Replicate 2

2.171

0.979

1.022

0.973

Replicate 3

1.824

1.218

1.007

0.923

  • The level of S100A7 mRNA expression for figure 6A was showed in table 2.

Table 5. The level of S100A7 mRNA expression in gMECs after treatment for 6 h

Replicates

Control

LPS(5 μg/mL)

0.1 μM E2

1 μM ICI

0.1 μM E2+ 1 μM ICI

0.1 μM G1

1 μM G15

0.1 μM E2+ 1 μM G15

Replicate 1

1.00

2.363

1.796

1.073

1.186

2.205

0.939

1.037

Replicate 2

1.284

2.265

1.646

1.111

1.170

2.537

0.791

1.336

Replicate 3

1.086

2.626

1.650

1.050

1.212

2.209

1.220

1.160

  • The concentration of S100A7 secretion in gMECs for figure 6B was showed in table 3.

Table 6. The concentration of S100A7 secretion in gMECs after treatment for 6 h

Replicates

Control

LPS(5 μg/mL)

0.1 μM E2

1 μM ICI

0.1 μM E2+ 1 μM ICI

0.1 μM G1

1 μM G15

0.1 μM E2+ 1 μM G15

Replicate 1

19.09

32.66

34.21

21.11

21.52

28.44

22.77

28.83

Replicate 2

20.12

32.86

33.21

20.47

29.10

27.88

21.06

20.10

Replicate 3

20.44

33.85

35.42

19.32

22.66

28.58

22.53

25.54

Reference:

1         Erskine RJ, Eberhart RJ, Hutchinson LJ, Spencer SB, Campbell MA: Incidence and types of clinical mastitis in dairy herds with high and low somatic cell counts. J Am Vet Med Assoc 1988;192:761-765.

2         Burton JL, Madsen SA, Yao J, Sipkovsky SS, Coussens PM: An immunogenomics approach to understanding periparturient immunosuppression and mastitis susceptibility in dairy cows. Acta Vet Scand 2001;42:407-424.

3         Madsen SA, Weber PS, Burton JL: Altered expression of cellular genes in neutrophils of periparturient dairy cows. Vet Immunol Immunopathol 2002;86:159-175.

4         Burvenich C, Paape MJ, Hill AW, Guidry AJ, Miller RH, Heyneman R, Kremer WD, Brand A: Role of the neutrophil leucocyte in the local and systemic reactions during experimentally induced E. coli mastitis in cows immediately after calving. Vet Q 1994;16:45-50.

5         Houdijk JG, Kyriazakis I, Jackson F, Huntley JF, Coop RL: Is the allocation of metabolisable protein prioritised to milk production rather than to immune functions in Teladorsagia circumcincta-infected lactating ewes? Int J Parasitol 2003;33:327-338.

Author Response

We would like to thank the reviewer for their accurate and detailed revision of our manuscript. Reviewer offered helpful and constructive suggestions and as a result, we feel that the manuscript has been significantly improved. We have seriously thought about the suggestions and provided our responses in a point-by-point manner.

Comment: In this manuscript, the authors investigated the role of estrogen in regulating the expression and secretion of S100A7 in goat MECs. The results showed that the S100A7 level is positively correlated with estrogen in goat milk, and the expression and secretion of S100A7 was up-regulated with estradiol treatment in goat MECs. This regulatory process was realized through activation of ERK signaling pathway. Therefore, the author concluded that estrogen activated ERK signaling pathway to induce the expression and secretion of antimicrobial peptide S100A7. This study is reasonably designed and had certain significance for understanding the specific role of estrogen in regulating the expression and secretion of antimicrobial peptides S100A7. However, there are several unclear questions need to be addressed before consideration for publication.

Response: Thanks for reviewer’s positive comments and advices of the work.

Major points:

Point 1: ELISA of materials and methods, page 3 Line 136.In this experiment, according to my understanding, the authors used ELISA to measure concentration of milk and cell supernatant, but the sensibility and specificity of ELISA weren’t showed, please added it.

Response 1: Thanks for reviewer’s kindly reminding. The limits of quantification of Goat S100A7 ELISA kit we used is 2.5 μg/mL - 80 μg/mL, the sensibility is less than 0.1 μg/mL; The limits of quantification of Goat estrogen ELISA kit we used is 3.2 pg/mL – 80 pg/mL, the sensibility is less than 1 pg/mL.

Minor points:

Point 1: Page2 Line 86-92: Please clearly mark the product code of the reagent used. And other reagents in materials and methods.

Response 1: Thanks for reviewer’s kindly reminding and advices. We have added the product code of the reagent used in the manuscript.

Point 2: Western blot, Page3 Line 130: In line 130, the authors used “Western blot”, while in line 192 and others showed “western blotting”, so western blotting need to revise into “Western blot” in the whole manuscript.

Response 2: Thanks for reviewer’s kindly reminding and advices. We have revised the “western blotting” into “western blot” in the whole manuscript.

Point 3: 3. Using “with estradiol treatment” instead of “with estradiol” in line 317.

Response 3: Thanks for reviewer’s kindly advices. We have used the “with estradiol treatment” instead of “with estradiol” in line 327.

Point 4: 4. Erβ need to revise into “ERβ” in line 319.

Response 4: Thanks for reviewer’s kindly advices. We have used the “ERβ” instead of “Erβ” in line 329.

Point 5: In Figure 1, the font of the ordinate in figure D is different from the font of the other figures, please unified.

Response 5: Thanks for reviewer’s kindly reminding and advices. We have unified the style of figure 1.

Point 6: In Figure 5B, the icon “control” change to “Control”. And add the size of the

protein to the right of the WB band, the protein name change to left side.

Response 6: Thanks for reviewer’s kindly reminding and advices. We have used the “Control” instead of “control” in figure 5B, added the size of the protein to the right of the WB band, the protein name was changed to the left side.

Point 7: Change the p value of significance to uppercase and italic in manuscript.

Response 7: Thanks for reviewer’s kindly reminding and advices. We have revised “p” into “P” in the whole manuscript.